# The Effects of Comprehensive Sexual and Reproductive Health/Family Planning Intervention Based on Knowledge, Attitudes, and Practices Among the Domestic Migrant Population of Reproductive Age in China: A Randomized Community Study

**DOI:** 10.3390/ijerph17062093

**Published:** 2020-03-21

**Authors:** Shuang-Fei Xu, Jun-Qing Wu, Chuan-Ning Yu, Yu-Yan Li, Rui Zhao, Yi-Ran Li, Ying Zhou

**Affiliations:** 1School of Public Health, Fudan University, Shanghai 200035, China; 19111020007@fudan.edu.cn; 2NHC Key Lab. of Reproduction Regulation (Shanghai Institute of Planned Parenthood Research), Fudan University, Shanghai 200035, China; lyy1033@163.com (Y.-Y.L.); zhaorui821030@163.com (R.Z.); llyyyrr@sina.com (Y.-R.L.); yingzhou2012@163.com (Y.Z.); 3Department of Chronic Disease, Longhua District Center for Chronic Disease Control/Mental Health, Shenzhen 510080, China; yu.chuanning@icloud.com

**Keywords:** China, family planning, intervention, domestic migrant population, sexual and reproductive health

## Abstract

*Background*: Domestic migrant populations are highly mobilized at a sexually active age, and often fail to meet their needs for contraception. Moreover, they assume sexual and reproductive health (SRH) risks and utilize fewer family planning services. *Method*: A quasi-experimental trial (community intervention) was adopted. Two-stage stratified cluster sampling was applied to recruit participants in Beijing and Chongqing. A comprehensive SRH/family planning intervention was implemented from August 4 2014 to August 3 2015. Propensity score matching (PSM) and multivariate probit models were adopted. Results: In total, 2100 and 2024 eligible participants were involved, and 815 and 629 pairs were matched by PSM in Beijing and Chongqing, respectively. The knowledge and attitudes of the participants regarding SRH and contraception were significantly improved through the comprehensive intervention. Reversible contraceptive methods were the most prevalent; couples largely decided to utilize condoms and family planning services. *Conclusions*: The comprehensive intervention had positive effects on knowledge, attitudes, and practices (KAP) for SRH/family planning among the domestic migrant population. The results acquired can be extrapolated to some extent, and the pattern of this intervention is well geared toward other similar settings in China.

## 1. Introduction

An official report released by the Chinese National Health and Family Planning Commission indicated that the domestic migrant population has been growing rapidly, reaching 247 million and making up 18% of the overall population in 2015 [1]. The sixth national population census indicated that, among the floating population, the sex ratio (male/female) was 1.29. More than 80% of the migrants were aged between 15 and 49 years old (especially 20–44 years old (70.6%)), 71.1% were educated below a junior high school level, and 74.7% migrated for business or to seek jobs [2]. However, this population is highly mobilized at a sexually active age and often fails to meet its needs for contraception. Additionally, migrants know less about reproduction and contraception than the general population and infrequently utilize family planning services: 17%~56% of the domestic migrant population aged above 18 years practice pre-marital sex [3,4,5,6], most of the participants scored under 30 on their knowledge of contraception, and over 50% of the participants could not answer how to correctly prevent Human Immunodeficiency Virus (HIV)/Acquired Immune Deficiency Syndrome (AIDS) [7,8,9]. Another study indicated that, compared to local residents, AIDS-related knowledge among the floating population was lower (63.1% vs. 57.9%) [10]. To address the unsatisfied needs for sexual and reproductive health (SRH) in this group, China has been striving to facilitate the SRH of the overall population, especially among the domestic migrant population, through various SRH intervention activities.

A health intervention describes an act performed for, with, or on behalf of a person or population who seeks to assess, improve, maintain, promote, or perfect their health, functioning, or health conditions [11]. Through a health intervention, people can acquire sufficient knowledge about contraception, HIV/AIDS prevention, and safe sexual behaviors [12,13]. By launching a reproductive health intervention and offering services to unmarried migrant women, people’s attitudes towards, and condom use behavior can be changed, and unintended pregnancies can also be effectively prevented [14]. As a previous study on reproductive health indicated that in Vietnam, providing knowledge on SRH and establishing better SRH behaviors through the intervention of prevalent instant communication directed toward domestic migrant women were of crucial significance [15]. As a study conducted by Mendelsohn in Shanghai uncovered, the participants were willing to accept a short-term educational intervention, which was conveniently accessible by the domestic migrant population [16]. A family planning intervention can effectively enrich knowledge of contraception and promote the utilization of family planning services among the domestic migrant population at a reproductive age [6,9]. The progress of these services in China, however, has been rarely evaluated in terms of the utilization of SRH/family planning services. SRH/family planning services among the domestic migrant population were only evaluated in one study conducted in Shenzhen, and the conclusion was reached that SRH services cover various districts in different ways [17]. On average, 60% of the participants had ever heard about the related policies or service information; less than 50% received the free SRH examination services. The study also indicated that 75% of the participants were satisfied with their SRH services.

In our research, a randomized community study with a comprehensive SRH/family planning intervention was conducted to increase the knowledge, improve the attitudes, and develop better practices for SRH/family planning among the migrant population.

## 2. Materials and Methods

### 2.1. Sampling Methods

Two-stage sampling was adopted. In the first stage, Beijing and Chongqing were selected as the study sites because they contain the largest domestic migrant population in North and Southwest China. In the second stage, cluster sampling was adopted in each city to sample four factories, construction sites, and entertainment venues in the streets where the domestic migrant population is concentrated. Local, well-trained investigators fully introduced the program details to the migrant populations at these sites. Based on informed consent, the floating population volunteered to participate in this study. Local investigators and coordinators at each site collected lists of participants and determined the eligible floating population using the inclusion criteria of ages between 15 and 49, not being registered as permanent residents in the cities where they were working, having lived and resided in these cities for at least half a year, and volunteering to participate in this study.

### 2.2. Sampling Size

We used the following formula to calculate the sample size:(1)$$n=\mathrm{Df}\;\times \;[1/(1-f)]\;\times \;\frac{{\left\{z_{\alpha }\sqrt{2p\left(1-p\right)}+z_{\beta }\sqrt{p_{1}\left(1-p_{1}\right)+p_{2}\left(1-p_{2}\right)}\right\}}^{2}}{{\left(p_{1}-p_{2}\right)}^{2}}$$
where Df denotes the effect of the sample design, recommended by the World Health Organization (WHO) as 1.5; f indicates the rate of the lack of a follow up, assumed to be 20%; z_α_ and z_β_ are 1.96 and 1.28, respectively; p_1_ and p_2_ are the rates of 50% and 60% for the knowledge of contraceptive methods before and after the intervention, respectively [4]; and p is defined as (p1 + p2)/2. The sample was calculated as 970 in each city. In the two cities, the total sample size reached 3880 (970 × 2 (the control and intervention groups) × 2 (number of cities)).

### 2.3. Study Design

A quasi-experimental trial (community intervention) was adopted. The factories, construction sites, and entertainment venues were split into control and intervention groups in different streets and the distance between each control group and intervention group was at least three kilometers to avoid “intervention contamination”. Each group involved two factories, two construction sites, and two entertainment venues. The intervention lasted for 12 months, from 4 August 2014 to 3 August 2015. The baseline investigation was completed in June 2014. Before and during the intervention, this intervention program was ensured to be exclusively conducted in the targeted sites by coordinating with relevant departments, which controlled for co-intervention bias. We made efforts to minimize follow-up loss, as the participants had signed long-term (three years or more) contracts with their employers (study sites), and this program lasted for only 12 months. In total 86 and 172 participants withdrew in Beijing and Chongqing, respectively.

### 2.4. Intervention Strategy

A professional working team was organized. This team was composed of ten providers of SRH services working at the study sites. The staff in the teams received standardized training courses. The courses lasted for four weeks. In the final round, all the trainees were required to take written and field examinations. Training certificates were issued to those who passed the exams.

Routine SRH/family planning services were offered to the control groups, including providing contraceptives, intrauterine device (IUD) insertion and removal, sterilization operations, abortion, and medical examinations according to the related policies and regulations.

For the intervention groups, SRH/family planning intervention was comprehensively carried out, including education on SRH/family planning, comprehensive counselling, technical support, and a follow-up (Figure 1). The intervention’s effects were evaluated from three perspectives: knowledge, attitudes, and practices (K.A.P), which together reflect the respondents’ understanding of the topic, their feelings and preconceived ideas towards it, and the ways in which they demonstrate their knowledge and attitude toward it through their actions [18].

(a)SRH/family planning education materials were developed to introduce the participants to the family planning regulations, as well as their rationales and applications, the pros and cons of contraceptive methods, knowledge about sexually transmitted diseases (STDs), and HIV/AIDS. The participants were given leaflets/brochures at least three times a month. The posters were put up in the specified areas and updated once a month. SRH lectures for the participants were held in the assigned places twice a month.(b)Counselling rooms were established to offer the participants counselling services for SRH/family planning at the study sites. SRH/family planning hot lines were also provided. Counselling files for in-depth analyses were generated to support the sustainable implementation of the intervention on the field. Expert counselling sessions were held once a quarter. The sessions could be held more frequently depending on the participants’ needs.(c)Technical services comprising maternal examinations, IUD examinations, and handling and checking certificates of marriage and childbirth for domestic migrant population (which are provided by the local department of family planning to prove the floating population’s identity, marital status, and birth status, and to facilitate the utilization of family planning services); were provided for free to the domestic migrant population. Contraceptives (pills and condoms) were also distributed. The participants were assisted by the working teams to select proper contraceptive methods in line with their own health conditions and encouraged the participants to take some novel contraceptives, including female condoms, IUDs and implants, etc. The teams encouraged the participants who were likely to engage in high-risk sexual behaviors to use condoms constantly. Actions were launched with several national welfare programs, such as the “cherishing girls action” (formulated by the State Council, which aims to protect the legitimate rights and interests of girls and to promote women’s development and gender equality) to facilitate the family planning benefit-oriented mechanism.(d)The follow-up was facilitated to be more pertinent, diverse, and standardized, and the follow-up quality and public service capacity were improved according to the requirements of “The Equalization of Family Planning Public Services for Migrant Populations”. A standardized and periodical follow-up was conducted by the working teams for the participants adopting contraceptive methods.

### 2.5. Quantitative Data Collected

The data for the intervention evaluation are presented in Appendix B: Table A1. The characteristics of the participants are listed in Appendix B: Table A2. 

### 2.6. Data Collection

At the end of the intervention, the participants were interviewed by the working teams in designated places of the study sites, such as conference rooms, dormitories, and dining halls. The interviewers (who were of the same gender as their respective interviewees) were charged with completing the questionnaires in one-on-one in-person interviews. The study coordinators evaluated the completeness and logic of the questionnaires, providing feedback on errors to the investigators when unqualified questionnaires were found. The responses to these questionnaires were then promptly revised by the interviewers with the interviewees.

### 2.7. Data Analysis

The data from all the questionnaires were assessed twice by different professionals using EpiData 3.1 (The EpiData Association, Odense, Denmark) to compare the data. Data cleaning included consistency verification for all variables. The analysis was performed in SAS version 9.3 (SAS Inc, Cary, NC, USA) and Stata MP version 14.0 (StataCorp, TX, USA). The frequencies and proportions were included in the descriptive statistics. Propensity score matching (PSM) was adopted to evaluate the net effects of the intervention. A sensitivity analysis for PSM was used to test the assumption of strongly ignorable treatment assignments [19]. Multivariate probit models were required to verify the results gained from PSM for dependent variable selection. Two types of biases were detected through the PSM sensitivity analysis and multivariate probit models for the unmeasured bias and the selection bias (Appendix A).

### 2.8. Ethics Approval and Consent to Participate

The study protocol was approved by the research ethics committee of the Shanghai Institute of Planned Parenthood Research (code PJ2014-20) before the program was implemented. All eligible participants were told the procedures of the study, with interpretation and clarification provided as required. Before the data collection, verbal and written informed consent forms were obtained from all participants to ensure the security and privacy of the information. For those aged between 15 and 17 who had migrated with their parents/guardians, their parents/guardians were told the details of consent and asked to sign the forms with the help of the community service providers. For those aged between 15 and 17 who migrated without their parents/guardians, the participants themselves signed the forms, accompanied by community service providers. Each of the field investigators signed a confidentiality agreement to protect the privacy and sensitive information of the interviewees.

### 2.9. Patient and Public Involvement 

There were no patients or public participants in this study.

## 3. Results

### 3.1. Comparison of the Characteristics between the Intervention and Control Groups in the Two Cities

In Beijing and Chongqing, 2186 and 2196 participants were recruited, respectively. During the study, 86 and 172 were lost to follow up, respectively. The main reason for a lack of follow-up was a job change. In the final analysis, 2100 and 2024 eligible participants were involved (Figure 2; Appendix B: Table A3, Table A4, Table A5, Table A6, Table A7 and Table A8).

“Group” was a dichotomous variable classified into “control” and “intervention”. It was included into a logistic regression model as the independent variable, and the variables on characteristics were also involved as the dependent variables in this model. The propensity score (PS) was estimated by the logistic regression model based on a probability given conditions. Appling the caliber method, the control and intervention groups were matched 1:1 by the PS. In the final analysis, 815 and 629 pairs were matched in Beijing and Chongqing, respectively. The differences before and after matching were compared by calculating the standardized differences (Appendix B: Table A3, Table A4, Table A5, Table A6, Table A7 and Table A8).

### 3.2. Participation in the Comprehensive Intervention among the Participants in the Two Cities

In Beijing and Chongqing, 92.70% and 38.95% received leaflets or brochures at least once, respectively, followed by lectures on SRH/family planning (Beijing: 45.44% and Chongqing: 36.59%). The interventions were participated in through diverse approaches, including watching video compact discs (VCDs), browsing posters, receiving face-to-face counselling, and engaging in counselling by phone (Table 1).

### 3.3. Effects of Intervention on Knowledge among Participants

In Beijing, 77.85% (819/1052) and 91.06% (958/1052) of the participants in the intervention group scored over 60 in terms of their knowledge on contraception and SRH, respectively, marking an increase of 47.12% and 33.52% compared with those in the control group (30.73% (322/1048) and 57.54% (603/1048), respectively). In Chongqing, 37.62% (336/973) and 65.16% (634/973) of the participants in the intervention group scored over 60 in terms of their knowledge on contraception and SRH, respectively, marking an increase of 28.77% and 25.29% compared with those in the control group (8.85% (93/1051) and 39.87% (419/1051), respectively). As the scores were normalized by rank transformation, the results of paired-t tests uncovered that the average ranked scores in the intervention groups were significantly higher than those in the control groups (p < 0.001) among the two cities. The results of the sensitivity analysis thus indicated that the assumption of a strongly ignorable treatment assignment was not rejected (Table 2).

### 3.4. Effects of the Intervention on Attitude and Practice among Participants

Across the two cities, in terms of the attitudes toward “what type of contraceptive methods do you expect to use?”, “is knowledge/information on SRH/family planning adequate?”, and “should men be involved in SRH/family planning education?”, the results of the McNemar tests indicated that the proportions of the attitudes in the intervention groups were significantly higher than those in the control groups (*p* < 0.05). The intervention had positive effects on these attitudes in the multivariate probit models (*p* < 0.05) (Table 3 and Table 4).

Among the two cities, for “what contraceptive methods are you adopting currently?”, “who determines the utilization of contraceptive methods?”, “have you received an IUD assessment service?”, and “have you used condoms for the last three sexual encounters?”, the results of McNemar tests revealed that the proportions of these practices in the intervention groups were significantly higher than those in the control groups (*p* < 0.05). For the practices involving utilization of SRH/family planning services, “have you gotten a ‘Certificate of Marriage and Childbirth for Domestic Migrant Population’?” and “have you participated in family planning services?”, the results of the McNemar tests showed that the proportions of the practices in the intervention groups were significantly higher than those in the control groups (*p* < 0.05). The intervention had positive effects on these practices based on the multivariate probit models (*p* < 0.05) (Table 3 and Table 4).

## 4. Discussion

When it comes to the KAP approach, education strategies for individuals and groups are needed to encourage positive practices and to avoid negative health behaviors. This approach is also dependent on comparatively unbiased information [20]. As the results of the intervention evaluation indicate, our intervention effected the KAP of SRH/family planning positively.

Men were encouraged to engage in SRH/family planning education. As a previous study about male SRH indicated, SRH has been traditionally focused on females. There are specialized health departments for women at all levels (from national to local). For instance, in Chongqing, few health settings and activities for SRH services are provided for males [21]. SRH/family planning services for men were, accordingly, indicated to be inadequate.

The proportions of couples adopting reversible contraceptive methods and couples deciding to use contraception were significantly higher in the intervention group than in the control groups. These two results indicate that the intervention exerted positive impacts by disseminating information about informed choices to the participants. The domestic migrant population learned to select diverse contraceptive methods autonomously and voluntarily and conceive when appropriate. Simultaneously, the proportions of participants checking their IUDs were improved by nearly 20%, and the proportions of those using condoms consistently also improved by 10% through the intervention. The intervention made progress in the protection and promotion of the participants’ SRH health, which was essential for the domestic migrants to obtain their Certificates of Marriage and Childbirth for the Domestic Migrant Population [22]. We saw notable improvements in the intervention groups in the proportions of the participants who obtained their certificates. The proportions of the participants who received family planning services were still low, although those proportions increased significantly in the intervention groups (Beijing: 31.53%; Chongqing: 10.49%). This phenomenon could have been caused by the quality of the services, the expense of family planning services (the services are free of charge in family planning stations but out-of-pocket expenses in hospitals), and the participants’ time using the services according to our field survey.

In this study, we found a discrepancy in the proportion of those in the intervention groups who received leaflets between Beijing and Chongqing. This discrepancy is mainly because the management of the floating population at the Chongqing site was not as good as that at the Beijing site. In Chongqing, the participants did not care about the leaflets and thought we were engaging in some commercial activities like a shopping promotion, even though the investigators fully explained the purpose to them. However, overall, the acceptance of the leaflets/brochures and SRH/family planning lectures outperformed the other intervention approaches for the participants. Facilitated by their low environmental requirements and less time and energy investments for the participants, the leaflets/brochures were dispatched to the participants to promote their education anytime and anywhere. SRH-related experts, professional workers of family planning, and celebrities engaged in public welfare were invited to give popular, friendly, and engaging lectures. These lectures were attractive to the participants.

The strengths of this study are embodied by the following three points. First and foremost, the intervention program was conducted in two cities, and significant intervention impacts were obtained. Secondly, a scientific and comprehensive intervention framework was conceived. For the smooth implementation of our program, professional SRH/family planning providers were recruited into the working team. Meanwhile, full support was received from the local administrative departments in charge of managing the domestic migrant population. Thirdly, during the phase of the intervention evaluation, two key statistical methods were adopted to analyze the net effects of the intervention, which is uncommonly found in previous studies.

The limitations of this study were that continuous interventions failed to be conducted for those participants who lived outside the working sites, who had shifted or travelled, or who had returned to their hometowns during festivals and holidays. For these participants, their absence from the working sites shrunk the time in which they received the intervention.

## 5. Conclusions

The objectives of this paper were reported for the interventions, design features, evaluation methods, and field experiences that correspond to the differences in the interventions among the domestic migrant population. We found that the SRH/family planning comprehensive interventions in Beijing and Chongqing exerted significant effects on migrants’ KAP. Specifically, intervention allowed more of the floating population to acquire SRH knowledge and adopt reversible contraceptive methods, and convinced couples to use contraceptive methods, constantly use condoms, and utilize family planning services. The acquired results can be extrapolated to some extent, and the patterns of our intervention are well geared toward other similar settings in China.

## Figures and Tables

**Figure 1 ijerph-17-02093-f001:**
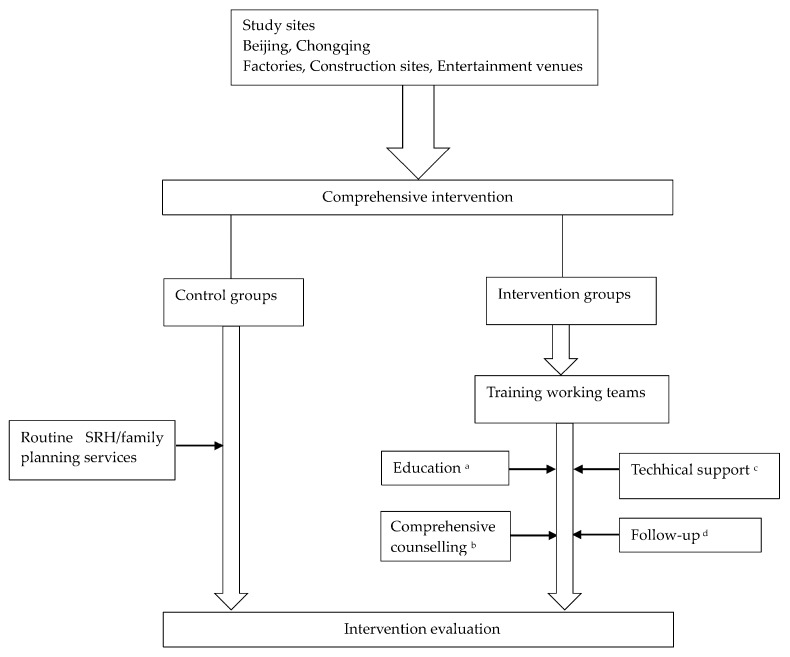
Implementation of the comprehensive intervention in the two cities.

**Figure 2 ijerph-17-02093-f002:**
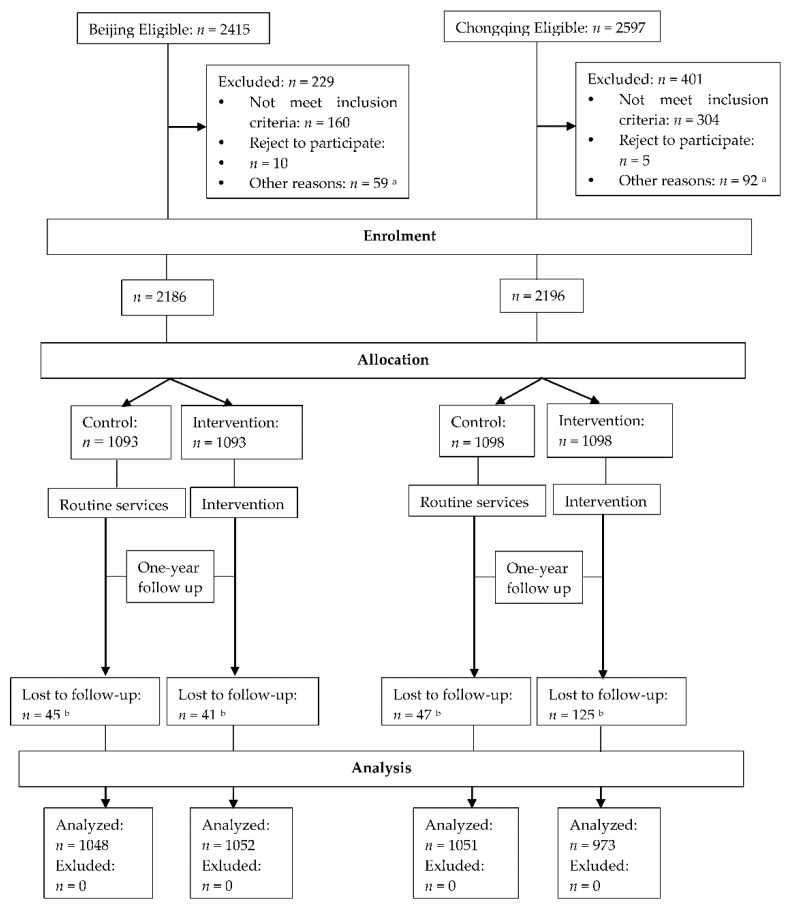
Flow of participants through each study stage in the two cities. ^a^ The primary reason was that the participants were absent in these sites. ^b^ Lost to follow-up was calculated as the total number allocated to the control and intervention groups, respectively, minus the number that received a follow-up at one year.

**Table 1 ijerph-17-02093-t001:** Participation in comprehensive interventions in the intervention groups of the two cities.

Variable	Beijing (*n* = 1052)		Chongqing (*n* = 973)	
	*n*	%	*n*	%
**Frequency of receiving leaflets/brochures**				
0	77	7.30	594	61.05
1~2	198	18.82	315	32.37
3~4	666	63.33	54	5.55
5~6	61	5.80	7	0.72
>6	50	4.75	3	0.31
**Interest in reading leaflets/brochures**				
Not interested	24	2.46	25	6.60
Interested in some of them	327	33.54	136	35.88
Interested in most of them	613	62.87	196	51.72
Interested in all	11	1.13	22	5.80
**Gains from leaflets/brochures**				
No gains	30	3.08	17	4.49
Having some gains	569	58.36	191	50.40
Having great gains	376	38.56	171	45.12
**Content assessment of leaflets/brochures**				
Too shallow	19	1.95	21	5.54
Too difficult	53	5.44	42	11.08
Moderately difficult	417	42.77	195	51.45
Very helpful	483	49.54	119	31.40
Other	3	0.31	2	0.53
**Frequency of participating in SRH/family planning lectures**				
0	574	54.56	617	63.41
1~2	280	26.62	321	32.99
3~4	140	13.31	28	2.88
5~6	40	3.80	6	0.62
>6	18	1.71	1	0.10
**Gains from SRH/family planning lectures ^a^**				
No gains	4	0.84	12	3.37
Having some gains	213	44.56	135	37.92
Having great gains	261	54.60	205	57.58
**Frequency of watching SRH/family planning VCDs**				
0	810	77.00	829	85.20
1~2	154	14.64	116	11.92
3~4	66	6.27	21	2.16
5~6	14	1.33	6	0.62
>6	8	0.76	1	0.10
**Gains from SRH/family planning VCDs**				
No gains	3	1.24	17	11.81
Having some gains	119	49.17	63	43.75
Having great gains	120	49.59	64	44.44
**Frequency of browsing posters**				
0	506	48.10	616	63.31
1~2	274	26.05	297	30.52
3~4	199	18.92	49	5.04
5~6	52	4.94	9	0.92
>6	21	2.00	2	0.21
**Interested in posters**				
Not interested	15	2.75	20	5.60
Interested in some of posters	290	53.11	153	42.86
Interested in most of posters	237	43.41	165	46.22
Interested in all	4	0.73	19	5.32
**Gains from posters**				
No gains	21	3.85	23	6.44
Having some gains	315	57.69	172	48.18
Having great gains	210	38.46	162	45.38
**Frequency of face-to-face counselling for SRH/family planning**				
0	740	70.34	717	73.69
1~2	229	21.77	227	23.33
3~4	65	6.18	18	1.85
5~6	18	1.71	10	1.03
>6	0	0	1	0.10
**Content assessment of face-to-face counselling**				
Too shallow	3	0.96	4	1.56
Too difficult	34	10.90	42	16.41
Moderately difficult	106	33.97	83	32.42
Very helpful	168	53.85	123	48.05
Other	1	0.32	4	1.56
**Frequency of counselling by phone**				
0	774	73.57	824	84.69
1~2	177	16.83	124	12.74
3~4	81	7.70	17	1.75
5~6	20	1.90	6	0.62
>6	0	0	2	0.21
**Content assessment of counselling by phone**				
Too shallow	3	1.08	5	3.36
Too difficult	29	10.43	24	16.11
Moderately difficult	103	37.05	52	34.90
Very helpful	141	50.72	66	44.30
Other	2	0.72	2	1.34

^a^. Four respondents who participated in the SRH/family planning lectures did not answer this question.

**Table 2 ijerph-17-02093-t002:** Effects of the intervention on the knowledge among participants.

Variable	Beijing (815 pairs)			Chongqing (629 pairs)		
	Control (Mean ± Std)	Intervention (Mean ± Std)	Sensitivity Analysis	Control (Mean ± Std)	Intervention (Mean ± Std)	Sensitivity Analysis
Knowledge on contraception	−0.61 ± 0.76	0.61 ± 0.77	t = 32.46, *p* < 0.0001; AIEI: (1.22, 1.23), IE = 1.23, adjusted 95%CI: (1.16, 1.31)	−0.37 ± 0.73	0.37 ± 0.99	t = 15.86, *p* < 0.001; AIEI: (0.74, 0.76), IE = 0.75, adjusted 95% CI: (0.65, 0.84)
Knowledge on SRH	−0.49 ± 0.87	0.49 ± 0.86	t = 22.78, *p* < 0.0001; AIEI: (0.98, 0.99), E = 0.99, adjusted 95%CI: (0.90, 1.07)	−0.37 ± 0.89	0.37 ± 0.95	t = 14.00, *p* < 0.0001; AIEI: (0.74, 0.76), IE = 0.75, adjusted 95% CI: (0.63, 0.85)

AIEI, adjusted intervention effect interval; IE, intervention effects; CI, confidence interval.

**Table 3 ijerph-17-02093-t003:** Net effects of the intervention on attitudes and practices among the participants by the McNemar test.

Variate (Quantitative Indicator)	Beijing			Chongqing		
	Intervention (%)	Control (%)	Sensitivity Analysis	Intervention (%)	Control (%)	Sensitivity Analysis
**Attitudes**						
	**713 pairs**			**410 pairs**		
What type of contraceptive methods do you expect to use? ** (% of “Reversible”)	94.25	89.48	S = 10.51, *p* = 0.0012, *p*-value interval: (0.0008, 0.0019) *	95.12	91.46	S = 4.59, *p* = 0.0321, *p*-value interval: (0.0213, 0.0329) *
	**815 pairs**			**629 pairs**		
Do you think about whether the knowledge/information for SRH/family planning is enough? (% of “Yes”)	60.12	47.36	S = 9.42, *p* = 0.0021, *p*-value interval: (0.0018, 0.0083) *	52.31	35.45	S = 39.01, *p* < 0.0001^#^*
Do you think about whether a man should be involved in SRH/family planning education? (% of “Yes”)	79.51	72.27	S = 11.72, *p* = 0.0006, *p*-value interval: (0.0005, 0.0021) *	73.29	59.30	S = 26.89, *p* < 0.0001^#^*
**Practices**						
	**713 pairs**			**410 pairs**		
What contraceptive methods are you using currently? ** (% of “Couples/sexual partners”)	95.23	92.14	S = 5.38, *p* = 0.0204, *p*-value interval: (0.0149, 0.0275) *	95.61	90.73	S = 7.41, *p* = 0.0065, *p*-value interval: (0.0038, 0.0067) *
Who determines the utilization of contraceptive methods? ** (% of “Reversible”)	94.59	89.74	S = 11.67, *p* = 0.0006, *p*-value interval: (0.0004, 0.0097) *	94.39	82.93	S = 25.39, *p* < 0.0001 **
	**114 pairs**			**115 pairs**		
Have you received an IUD assessment service? ** (% of “Yes”)	76.32	56.14	S = 9.61, *p* = 0.0019, *p*-value interval: (0.0005, 0.0010) *	74.78	57.39	S = 6.90, *p* = 0.0086, *p*-value interval: (0.0054, 0.0093) *
	**837 pairs**			**520 pairs**		
Have you used condoms in the last three sexual encounters? ** (% of “Yes”)	65.23	57.96	S = 9.37, *p* = 0.0022, *p*-value interval: (0.0018, 0.0085) *	52.31	39.81	S = 16.44, *p* < 0.0001 *
	**815 pairs**			**629 pairs**		
Have you gotten the “certificate of marriage and childbirth for domestic migrant populations”? (% of “Yes”)	78.40	73.25	S = 5.92, *p* = 0.0150, *p*-value interval: (0.0126, 0.0331) *	42.77	31.00	S = 18.88, *p* < 0.0001 *
Have you participated in for family planning services? (% of “Yes”)	31.53	22.94	S = 15.12, *p* = 0.0001, *p*-value interval: (0.0001, 0.0004) *	10.49	3.66	S = 22.83, *p* < 0.0001 **

*, the increase in odds was less than 5%; **, the selected sample; #, the upper and lower bounds of the *p*-value interval were too small to show.

**Table 4 ijerph-17-02093-t004:** Effects of the intervention on attitudes and practices among participants by multivariate models.

Variable	Beijing	Chongqing
	Models	Models
**Attitude**		
	***n* = 1864**	***n* = 1476**
What type of contraceptive methods do you expect to use? ^△^	Trivariate probit model^▲^^★^: intervention: (β = 0.24, p = 0.0129), 95%CI: (0.05, 0.43)	Trivariate probit model^▲^^★^: intervention: (β = 0.27, p = 0.0116), 95%CI: (0.06, 0.48)
**Practice**		
	***n* = 1864**	***n* = 1476**
What contraceptive methods are you adopting currently? ^△^	Trivarate probit model^▼^^★^: Intervention: (β = 0.25, p = 0.0159), 95%CI: (0.05, 0.46)	Trivariate probit model^▼^^★^: Intervention: (β = 0.47, p < 0.0001), 95%CI: (0.25, 0.69)
Who determines the utilization of contraceptive methods? ^△^	Trivarate probit model^▲^^★^: Intervention: (β = 0.37, p < 0.0001), 95%CI: (0.19, 0.54)	Trivarate probit model^▲^^★^: Intervention: (β = 0.53, p < 0.0001), 95%CI: (0.32, 0.75)
	***n* = 401**	***n* = 480**
Have you received an IUD assessment service? ^▽^	Quavarate probit model^▲^^★^: Intervention: (β = 0.51, p < 0.0001), 95%CI: (0.26, 0.75)	Quavariate probit model^▼^^★^: Intervention: (β = 0.46, p < 0.0001), 95%CI: (0.21, 0.71)
	***n* = 2077**	***n* = 1631**
Have you used condoms in the last three sexual encounters? ^☆^	Heckprobit mode^▲^^★^: Intervention: (β = 0.16, p = 0.0052), 95%CI: (0.05, 0.27)	Heckprobit mode^▲^^★^: Intervention: (β = 0.37, p < 0.0001), 95%CI: (0.22, 0.51)

★, PS was insignificant; ▲, no selection bias; ▼, there was a selection bias. △, Three-step selection was conducted to establish the trivariate-selected samples (the first selected sample included those who practiced sexual behavior; the second included those from the first sample who used contraception; the third included the information we were interested in from the second sample);.▽, Four-step selection was performed to establish equivariant-selected samples (the first two steps were identical to those on attitude, the third sample included those who used IUDs, and the fourth included the information we were interested in from the third sample); ☆, Heckprobit models were adopted (the first sample referred to those who engaged in sexual behavior; the second selected the factors we were interested in from the first sample).

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
