# Peer review of "The Effects of Comprehensive Sexual and Reproductive Health/Family Planning Intervention Based on Knowledge, Attitudes, and Practices Among the Domestic Migrant Population of Reproductive Age in China: A Randomized Community Study"

_ijerph, 2020, doi:10.3390/ijerph17062093_

Round 1

Reviewer 1 Report

Study results hold significance and relevance for adoption of family planning services and contraception usage among immigrant populations. But there are a lot of visible problems with the English language use in the paper and considerable number of typos (or possibly erroneous language usage).

Author Response

Dear reviewer, thanks for your comments. We are sorry for the English language. We hope the revised manuscript could be acceptable for you.

Reviewer 2 Report

Overall, the topic is important and has implications for improving sexual and reproductive health outcomes for migrant workers in China.  However, the way in which the manuscript is currently structured makes it difficult at times to appreciate the importance/significance of the findings.  I strongly encourage the authors to find an English as a first language speaker to help with many of the grammar, phrasing and sentence structure issues throughout the paper.  That will address some, but not all, of the sections where your meaning isn't 100% clear.

Below is section by section feedback.  Because of the numerous issues throughout the manuscript, the comments below are not meant to be exhaustive.

Abstract:

Line 19:  It should be either --"THE domestic migrant population is..." OR "Domestic migrant populations ARE highly..."

Line 19 (and later on in the paper):  What is meant by "active in sexual needs"?  That suggests that they have a strong sexual desire (strong need for sex), which may be true, but I don't think that that is what you mean.

Line 20:  Your meaning regarding "fails to satisfy the requirements on contraception" is unclear.  Also, whose requirements are you referring to?  

Line 31 (and elsewhere):  What does KAP refer to?

Introduction:

Line 38:  I don't think that the observation about migrants and China's economy is even needed.  You could start the paragraph with the observation about the growing migrant population.

Line 40:  Change "taking up..." to "MAKING up..."

Lines 42-42:  When you observe that the population knows less about reproduction and contraception, what is this based on?  Compared to whom?

Lines 44-46:  Provide more context for who was studied to get the data reported?

Line 55 (and elsewhere):  You use the word "bespoke."  Not sure what you mean, but that is not likely the proper term for whatever you are trying to say.

Line 57:  Authors are not referred to by full first name and last initial.

Line 64:  The conclusion seems to go beyond the fact that SRH services covered various districts.  Rather, the conclusion(s) seems to be the information that follows in the next 2-3 sentences.

Methods:

Line 93:  August - August is actually 13 months.  In comparison, for example, August 1, 2019 - July 31, 2020 would be 12 months.

Line 99:  How many withdrew from the study?

Intervention Strategy:

Line 107:  Does "artificial termination of pregnancies" mean abortion?  If so, say that.

Data Collection:

Lines 144-145:  Change "...face-to-face interviews with no other party" to something like -- one-on-one in person interviews.

Results:

In the flow chart, change "Reject to participate" to "Declined to participate."

Line 183 (and elsewhere):  Change "Among two cities" to "In the two cities" or "Across the two cities..."

Line 194 (and elsewhere):  "Basically" is too informal.

Line 195:  How can you state that all (or most) of the participants received leaflets when only 38.95% of them did in Chongqing?

Line 201:  Make it clearer closer to the start of the sentence which group represents the 77.85% and which the 91.06%.

Line 207:  I think that you mean "sensitivity" analysis.

Lines 234-235:  It is not clear what is meant by "...the proportions of the practices in the intervention groups were significantly higher" given the ways in which the questions are phrased.  What represents a higher response for "what contraceptive methods [are being] adopt[ed] currently"?  Does that mean that you had a preferred method that migrants should be using and that more intervention participants chose that method?  If so, which method was it?

Similar issue with the question "who determines the utilization of the contraceptive methods"?  Was there a preferred answer?  What was it?

Conclusion:

This section is very under-developed.

Author Response

Dear reviewer,

Thanks for your comments. I have made several modifications as your suggestions. The inappropriate or error sentences and words that you pointed out have corrected. Some specific questions were detailedly explained as following.

Introduction:

Lines 42- 46 were cited from the same lectures.

Line 64 was indeed the information that follows in the next 2-3 sentences.

Methods:

Line 93:  August - August isn’t always 13 months, like 2014/08/02-2015/8/01, 2014/08/24-2015/8/23. Now I specified the date.

Results:

Line 195: I stated all (or most) of the participants received leaflets because the majority of all participants (Beijing+Shanghai) received leaflets.

Lines 234-235: the preferred answers to each questions in table 3 and table 4 were indicated in appendix table A1. This time I have made some modifications in table 3. If you think it is necessary, I can remove the variable declaration tables from appendix to the main body.

Conclusion:

More information was added to the conclusion section.

Reviewer 3 Report

This study examines the impact of an intervention program including education and reproductive health services on sexual/reproductive health knowledge and behavior among domestic migrant young and middle adults in two urban areas in China. Strengths of this study include the use of high-impact intervention to address an important public health issue in a population that has been under-represented in global health research. I have a number of concerns about the clarity and completeness of the communication which I think need to be addressed.

The sampling strategy is not described in sufficient detail to fully evaluate its validity. The criteria for selecting the clustering sites is relatively clear, but there is no description of how individual participants were selected within clusters. It is important to describe how participants were recruited (by whom, in what setting, with what materials, etc.). The background regarding the migrant population in China is not described sufficiently for a global audience. More explanation is needed regarding the composition of this population, motivations for migration, and the reasons that members of this group are at elevated risk for sexual and reproductive health problems. Some terminology needs to be explained for a global audience. For example, it is not clear what the term “caring girls” (line 131) means or what a “marriage certificate for migrant populations” (line 236) is. The term “KAP” appears to refer to the intervention strategy, but I cannot find a definition in the paper. The results state that most of the participants in the intervention group received leaflets, but this appears to be the case on in the Beijing cluster; according to both the text (line 195) and Table 1, few than 40% of participants in the Chonqing intervention group reported receiving them (compared with 97% in Beijing). This discrepancy needs to be explained. There are substantial issues with English language grammar and usage throughout the manuscript that make many elements difficult to understand. The manuscript needs to be heavily edited for fluency.

Author Response

Dear reviewer,

Thanks for your comments. I have made several modification.

  1. describing the sampling strategy more clearer;
  2. enriching background regarding the migrant population in China as your suggestions;
  3. explaining the confusing terminologies;
  4. the discrepancy of the proportion of receiving leaflets in intervention groups between Beijing and Chongqing was mainly due to that the management of floating population in Chongqing was not as good as that in Beijing. In Chongqing, participants didn’t care about the leaflets and thought we were doing some commercial activities like shopping promotion, though the investigators fully explained purpose to them.
  5. We are sorry for the English language, and hope the revised manuscript could be acceptable for you.

Round 2

Reviewer 2 Report

The manuscript has a wide range of issues that need to be addressed before publication.  Below is an overview of some such issues, section by section.  I continue to encourage the authors to have the manuscript reviewed by someone proficient in the English language.

Abstract:

Of the four previous comments I made for this section, only one has been adequately addressed.  The first and third comments still need to be worked on.  And the new phrasing -- "stay in sexually active stage" is both awkwardly phrased and unclear in its meaning.

Introduction:

Line 43 -- What is meant by "floating population"?  

Line 46 -- Both "stay in sexually active stage" and "fails to satisfy their requirements" are awkwardly phrased, with their meaning unclear.

Lines 48-49 -- That 17% engaged in premarital sex does not seem like a significant percentage of a population to warrant concern.

Line 51 -- Based on how the previous sentences are structured, it is not 100% clear what "the problem" is.  

With that said, the problem I believe you are building up to is that migrant populations are particularly in need of SRH services.  However, the way (and order) in which the information is presented makes it difficult to come to that conclusion.  For example, how does the migrant population compare with the native population in terms of risks?  You note that 17% of the former engage in premarital sex, how does that compare to the latter population?  If 50% of the former could not answer correctly questions re: preventing HIV/AIDS, how does that compare to the native population?

It is not sufficient to list a number of characteristics of the migrant population without providing sufficient context.

Line 52 -- Change "whereby" to "through."

Line 62 -- Continued issues with how authors are referred to.  Full names are typically not used.

The last two paragraphs (the last one which is only a sentence long) need a lot of work.  The order in which information is presented, sentence structure issues, and too brief a description of the intervention are just some of the issues that need to be addressed.

Materials and Methods:

Line 108 -- Here or elsewhere, describe the main reasons for the participants that were lost to follow-up.

Line 137 -- Why is the checking of marriage certificates relevant to SRH/family planning?

Lines 142-143 -- Unclear what is meant by "Cherishing girls action" and "Pairing support of only-child families in financial difficulty."

Lines 156-158 -- I don't think that you mean that questionnaires were revised, rather that responses to the questionnaires that didn't make sense were revised.

Line 182 -- Although the participants may not have been patients seen in hospitals or clinics, weren't they assisted with such things as IUD placement, when indicated?  

Results:

Lines 194 - 200 -- Here and elsewhere sentence structure needs a lot of work as your meaning is not 100% clear.

Line 205 -- Here and elsewhere, it should be:

Among the two cities, In the two cities, Across the two cities OR name both cities at the start of the sentence.

Line 206:  Both I and one other reviewer noted that 38.95% is NOT "most participants."  In fact, that is a significant difference from the percentage reported for Beijing.

Lines 212-217 -- Needs a lot of work.  It seems that you're saying that 77.85% of those in the intervention group scored above a 60 on contraception knowledge and 91.05% scored above 60 on SRH and that that was an increase over their baseline scores.  But when you note that these scores were better than those of the control group, what were their scores or what percentage scored over 60?

Similar issue with the way that the results for Chongqing are presented.

Line 224 -- It is still not clear to me what a "higher" score means for questions such as "what type of contraceptive methods you expect to use?".

Lines 249-252  - the relevance of Certificates of Marriage should be described the first time they are mentioned.

Discussion:

Line 290 -- More attractive than what?

Line 291 -- What is meant by "one point to multiple points"?

Line 293 -- Why is it significant that the intervention was conducted in two cities?  Explain why this matters.

Conclusion:

Line 308 -- Is Shanghai the same thing as Chongqing?

Author Response

Dear reviewer, 

Thanks a lot for your suggestions. Hope the revised version will be accepted by you.

Reviewer 3 Report

Thank you for addressing a number of my concerns regarding the initial draft of the manuscript. I have two remaining comments:

  1. The manuscript still needs to be extensively edited for correct use of English. You should consult with a collaborator who is a native English speaker, or with a professional editing service to revise your manuscript for fluency and readability.
  2. I appreciate your response to point #4 in your letter, regarding the discrepancy in reported exposure to the intervention between sites, but I do not see this addressed in the manuscript itself. It is important to include this in the discussion section, as understanding the reasons for these differences may be critical to anticipating whether the same intervention is likely to be successful if applied in other areas.

Author Response

Dear reviewer,

Thanks a lot for your help. We have added the explanation for discrepancy of the proportion of receiving leaflets in intervention groups between Beijing and Chongqind in disscussion part.

Hope the revised version will be accepted by you.